# Meningioangiomatosis Combined with Calcifying Pseudoneoplasms of Neuraxis

**DOI:** 10.3390/brainsci13050786

**Published:** 2023-05-11

**Authors:** Xiangyu Sun, Chengshi Xu, Yuxiang Cai, Zhiyong Pan, Zhiqiang Li

**Affiliations:** 1Department of Neurosurgery, Zhongnan Hospital of Wuhan University, Wuhan 430062, China; 2021283030164@whu.edu.cn (X.S.); xucs1985@163.com (C.X.);; 2Department of Pathology, Zhongnan Hospital of Wuhan University, Wuhan 430062, China; muyangcyx@whu.edu.cn

**Keywords:** meningiomatosis, calcifying pseudoneoplasms of neuraxis, rare, histopathology

## Abstract

Meningioangiomatosis (MA) is a rare hamartomatous or meningovascular lesion involving the central nervous system, and is sometimes associated with intracranial meningiomas. Calcifying pseudoneoplasms of the neuraxis (CAPNON) are rare, slow-growing benign tumor-like lesions that can occur anywhere along the neuraxis. Here, we report a rare case of MA combined with CAPNON. A 31-year-old woman was admitted to our hospital because of a high-density mass in the left frontal lobe, detected by computed tomography (CT) during a physical examination. She had a 3-year history of obsessive–compulsive disorder. We describe the imaging, histopathological, and molecular characteristics of the patient. To our knowledge, this is the first report describing MA combined with CAPNON. We reviewed the literature on MA and CAPNON over the last decade and summarized the points for differential diagnosis and treatment. It is difficult to preoperatively distinguish between MA and CAPNON. However, this coexisting condition should be considered when intra-axial calcification lesions are observed on radiological imaging. Accurate diagnosis and appropriate treatment are likely to benefit this patient group.

## 1. Introduction

Meningioangiomatosis (MA) is a rare hamartomatous or meningovascular lesion involving the central nervous system. It occurs sporadically, or may be associated with neurofibromatosis type 2 (NF-2). The sporadic symptoms are headaches, seizures, and a variety of other neurological symptoms. MA associated with NF2 is usually asymptomatic and found incidentally [1]. MA was first described by Bassoe and Nuzum in the autopsy of a patient with NF2 in 1915 [2]. Later, the brain lesion was named ”meningioangiomatosis” by Worster-Drought et al. in 1937 [3]. Calcifying pseudoneoplasms of the neuraxis (CAPNON) are rare, slow-growing, benign tumor-like lesions that can occur anywhere along the neuraxis, including the brain and spine [4,5]. Originally described by Rhodes and Davis in 1978, CAPNON is also called fibro-osseous lesions [6]. In this article, we describe a rare co-existence of MA and CAPNON that has never been previously reported.

The authors report a case involving a 30-year-old woman with a high-density mass in the left frontal lobe, suspected to be a meningioma on computed tomography (CT). However, after surgical resection (Operative video see Appendix A), the histopathological diagnosis of the lesion was MA combined with CAPNON. This study aims to provide a detailed analysis of this rare entity, including its clinical presentation and histopathological and imaging features, by collecting reports from other authors and our practical experience with a patient who underwent surgical resection at our institution. It is crucial to distinguish benign lesions from the more common vascular malformations and calcified vascular, neoplastic, or non-neoplastic differential diagnoses, because complete resection is curative [7]. In addition, we aim to analyze the possible mechanisms of its occurrence and development to provide references for treatment decisions.

## 2. Materials and Methods

We present a case of a patient with a histopathologically confirmed diagnosis of MA combined with CAPNON. Additionally, we conducted a comprehensive review of the literature on MA and CAPNON published in PubMed over the past 10 years. The search keywords included: meningioangiomatosis, calcifying pseudoneoplasm, and calcifying pseudotumor. The following parameters were collected from the qualifying articles: study type, number of patients, anatomical area, clinical presentation, radiological presentation, therapy, complications, and outcomes. In this review, we analyzed MA and CAPNON separately and discussed the mechanisms underlying their occurrence and development. This study was approved by the Ethics Committee of the Zhongnan Hospital of Wuhan University (no. 2019048).

## 3. Case Presentation

A 31-year-old woman was admitted for physical examination due to a high-density mass found by computed tomography (CT) in the left frontal lobe. The patient had a 3-year history of obsessive–compulsive disorder and experienced progressive improvement after taking sertraline tablets for 6 months. The CT scans (Figure 1A–C) revealed an irregular hyperdense mass in the left frontal lobe, measuring approximately 19 × 15 × 14 mm. Subsequent magnetic resonance imaging (MRI) revealed a hypo-intense mass with an unclear boundary on the T1-weighted image (Figure 1D) and an irregular mixed hypo-intense mass on the T2-weighted image (Figure 1E) and T2 FLAIR (Figure 1F). Irregular linear enhancement was observed on gadolinium-enhanced T1-weighted MRI (Figure 1G–I). Based on these initial images, meningioma was suggested.

The patient underwent a left frontotemporal craniotomy and complete resection. The gray-white and gray-brown lesions were relatively compact, and the boundary between the lesions and the surrounding brain tissue was unclear (Figure 2A). Immediate postoperative MRI confirmed the complete removal of the tumor. (Figure 1J–L). The patient recovered well, and no recurrence was observed on the MRI (Figure 1M–O) 3 months after surgery. However, the symptoms of the obsessive–compulsive disorder did not show considerable improvement.

## 4. Histopathology

Microscopic observation using hematoxylin and eosin (H&E) staining revealed that the lesion was primarily located between the pia mater and the cerebral cortex. A clear border was observed between the MA and CAPNON (Figure 2B,E). In the superficial cortical areas of the brain parenchyma, proliferating vessels surrounded by spindle cells were observed. Psammoma bodies, partially surrounded by small proliferating blood vessels, were visible, which are a classic pathological feature of MA (Figure 2B–D). Large calcifications were observed in the pia mater, and the calcified components were irregular or nested with proliferating spindle cells around them, which are typical signs of CAPNON (Figure 2E–G).

Immunohistochemically, the meningothelial cells positively expressed CD34 (Figure 2H), the somatostatin receptor (SSTR-2), the epithelial membrane antigen (EMA), vimentin (Figure 2I), and the progesterone receptor (PR) to varying extents. The stained meningothelial cells were mainly located on the surface of the pia mater and around the blood vessels of the cerebral cortex. Neurons invading the lesion were immunopositive for neuronal nuclei (NeuN), whereas the glial cells were immunopositive for glial fibrillary acid protein (GFAP) and oligodendrocyte line transcription factor 2 (Olig-2). The Ki-67 proliferation index was low (1%).

## 5. Results

Based on the results of H&E and immunohistochemical staining for specific biomarkers, the histopathological diagnosis of this lesion was MA combined with CAPNON, accompanied by the proliferation of meningothelial cells.

## 6. Discussion

MA and CAPNON are two rare lesions of the nervous system. We conducted a comprehensive review of the literature on MA and CAPNON published on PubMed in the last 10 years. Our review included 62 cases of MA (Table 1) and 50 cases of CAPNON (Table 2).

Among the 62 patients with MA, 43 were male (69.35%) and 19 were female (30.65%) patients, suggesting that MA has a male predominance. The ages of the 62 patients ranged from 8 months to 73 years, with an average of 19.85 years. However, 35 of the 62 patients were under 18 years of age (56.45%), indicating that MA has predominance in young patients. Most MA lesions are sporadic. From 2013 to 2022, NF-2 signs were only reported in 1 of the 62 MA cases in PubMed [8]. NF2 is a neurocutaneous disorder that may potentially develop into schwannomas, meningiomas, and ependymomas. A 2-year-old boy presented with multiple cystic meningioangiomatosis and a grade II ependymoma in the right cerebellum, which was found incidentally after trauma. In sporadic MA cases, the most common symptoms were refractory seizures (46/61; 75.41%) and/or headaches (9/61; 14.75%). Among the 62 MA cases, the lesions were mostly located in the temporal (22/62, 35.48%), frontal (21/62, 33.87%), and parietal lobes (16/62, 25.81%), and could be associated with epilepsy and mental symptoms. Only six cases occurred in the occipital lobe (9.68%). Intracranial lesions coexisting with MA have been reported in 17 patients [8,9,10,11,12,13,14,15], and meningioma was the most common combined tumor (11/17, 64.71%) [9,10,13,14,15]. Notably, the perivascular spread of meningioma-associated-type MA should not be interpreted as evidence of a grade II meningioma.

Among the 50 patients with CAPNON, 23 were male and 27 were female, with a mean age of 47.63 years and ranging from 8 to 73 years old. Although CAPNON can occur anywhere along the neuraxis, the intracranial location was the most common (39/50, 78%), with the remaining cases occurring in the spine (9 cases) and the craniocervical junction (2 cases). Furthermore, among the 39 intracranial CAPNON cases, 26 (66.67%) were supratentorial and 13 (33.33%) were infratentorial. Ho et al. reported that nearly one-third of CAPNONs were “collision” lesions, where the CAPNON tissue coexisted with a separate, distinct entity [16]. In seven cases, coexistent primary tumors were described, including meningioma [17], lipoma [18,19,20], and low-grade glioma [21]. The most common symptoms of intracranial CAPNON were headaches (17/39, 43.59%) and/or seizures (13/39, 33.33%), with a few patients presenting with cranial nerve defects, such as hoarseness, decreased hearing, or gait disturbance. In a 2013 study by Stienen et al. [7]., the authors reviewed all reported cases of CAPNON between 1977 and December 2011. Of the 22 patients with intracranial CAPNON, who had a mean age of 45 years, 19 had supratentorial CAPNON. The modes of presentation included epileptic seizures in eight patients, headache in five, cranial nerve affection in four, and dizziness and limb paresis in three. This finding is consistent with that of our literature review.

Diagnosing MA and CAPNON can be challenging due to the lack of characteristic signs to differentiate between them on preoperative imaging. MA often presents with a certain extent of calcification and contrast enhancement on CT and MRI [22]. One study reported that calcification was prevalent in 89.6% of the patients with MA. The lesion is generally confined to the cortex [23,24], with mostly low signals on MRI-T1WI and dark, wavy “cyclotron” signals on T2WI. In contrast, CAPNON shows little or no contrast enhancement with severe or peripheral calcification on CT and MRI. Due to the lack of specific clinical presentations and typical imaging features [25], postoperative pathological diagnosis is crucial for both MA and CAPNON. MA is generally characterized by proliferative meningothelial and fibroblast-like cells that infiltrate the leptomeninges with hypercellular areas and sclerosis [10], which majorly involves the outer layers of the cortex. In the adjacent cortex, large ganglion cells are isolated by thickened blood vessels, and the neuronal fibers in the trapped ganglion cells show tangling changes. The histological features of MA include meningovascular and leptomeningeal hyperplasia and calcification. Immunohistochemical staining is not very effective in diagnosing MA due to the lack of consistent markers [13]. The common histological features of CAPNON include chondromyxoid regions, palisading spindle cells, fibrous stroma, calcifications, and psammoma bodies [26]. However, the presence of these components is highly variable among the reported cases. The major immunohistochemical findings were the presence of epithelial membrane antigen (EMA) and vimentin, as well as the absence of glial fibrillary acid protein (GFAP) and S-100 protein [7]. As shown in Table 2, EMA was positive in 17 cases (70.83%; 17/24), and vimentin was detected in all 10 cases (100%; 10/10). However, the S-100 and GFAP were negative in 14 (82.35%; 14/17) and 11 cases (61.11%; 11/18), respectively.

The differential diagnoses of MA mainly include meningioma, vascular malformation, and cavernous hemangioma. Meningiomas are usually associated with dural diseases, whereas MAs often involve the cerebral cortical surface. Vascular malformations and cavernous hemangiomas generally lack perivascular meningothelial cell proliferation or psammoma bodies. In contrast, the most prominent feature of CAPNON is its highly dense calcification; therefore, oligodendrocytoma, astrocytoma, meningioma, osteosarcoma, chondrosarcoma, and neoplastic calcinosis are all important differential diagnoses.

Surgical resection is the most effective treatment for MA and CAPNON. In this study, 58 MA cases (93.55%;58/62) and 46 CAPNON cases (92%;46/50) were selected for surgery, particularly in high-risk areas. For instance, Lucila Domecq Laplace et al. [27] reported three cases of CAPNON located in the posterior fossa with perilesional edema, and the vast majority were cured after total resection. However patients may experience recurrence because of incomplete evacuation [28], neoplastic malignancy [12], or coexisting with other malignant tumors [20,21]. In addition to surgery, medical treatment and follow-up observations have been reported [1,29,30,31]. Shlomit et al. [29] reported a 67-year-old man with diffuse meningioma involving the occipital and right temporal lobes. After being diagnosed with MA by biopsy, the patient benefited from antiangiogenic therapy with bevacizumab, which led to clinical stability and marked radiological improvement. Similarly, indomethacin, an anti-inflammatory drug, was administered as a treatment option in a 44-year-old woman with chest CAPNON. After completing the treatment, the patient’s symptoms were completely relieved, and the lesion disappeared on the CT and MR imaging [31]. These successful non-surgical treatments shed light on the mechanisms underlying the development and progression of MA and CAPNON.

## 7. Conclusions

Herein, we report a rare case of MA coexisting with CAPNON, which was treated with tumor resection and had a good prognosis. Histological H&E staining revealed proliferating vessels and psammoma bodies encased by blood vessels, suggesting that the pathogenesis of MA is associated with cortical vascular malformations and the secondary proliferation of meningothelial cells. The formation of psammoma bodies may be stimulated by vascular proliferation. Most current studies suggest that CAPNON may represent a series of reactive processes that can arise in association with diverse underlying pathologies, including inflammatory, degenerative, vascular, and neoplastic lesions. In the present case, meningeal vascular proliferation and fine leptomeningeal epithelial proliferation may have led to the benign reactive lesions of CAPNON. Although nonsurgical treatment has been successful in some cases, complete resection is still attempted, when feasible, to relieve the patient’s symptoms and perform histopathological analysis. Definitive diagnosis is based on histopathological analyses to prevent the patient from having to undergo aggressive adjuvant therapy.

## Figures and Tables

**Figure 1 brainsci-13-00786-f001:**
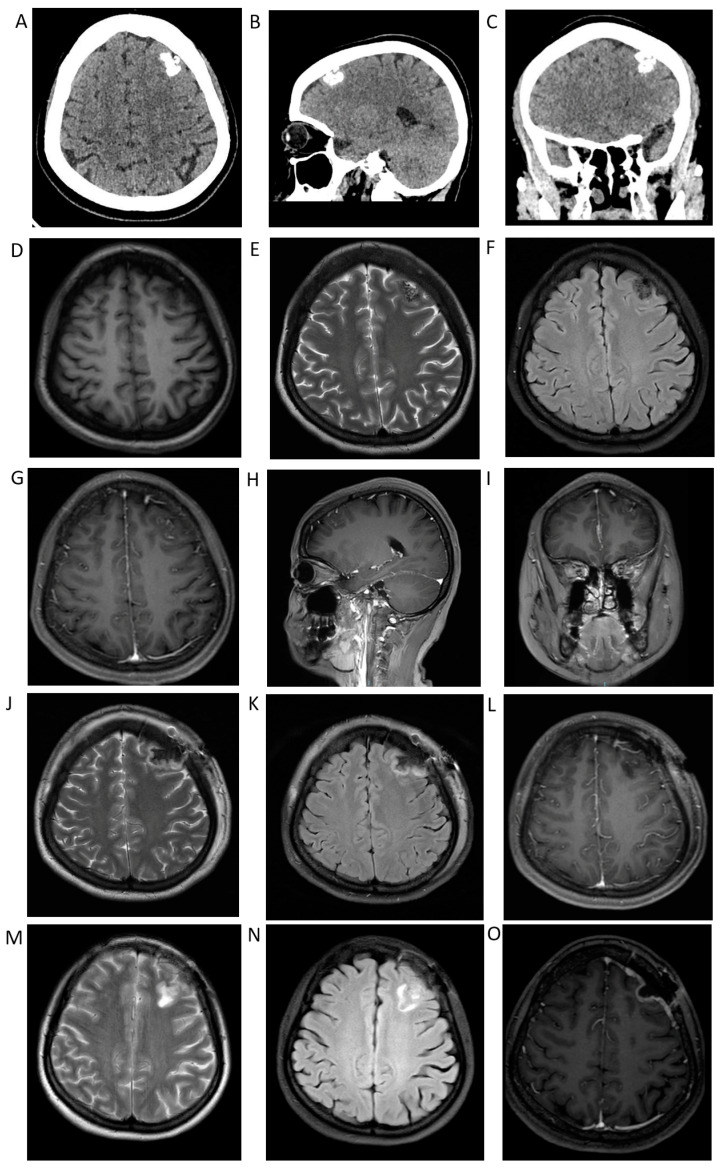
(**A**–**C**) Computed tomography (CT) shows an irregular hyper-dense mass in the left frontal lobe, measuring approximately 19 × 15 × 14 mm. Subsequent magnetic resonance imaging (MRI) demonstrates (**D**) a hypo-intense mass with an unclear boundary on the T1-weighted image and (**E**,**F**) an irregular mixed hypo-intense mass on the T2-weighted image and T2 FLAIR. (**G**–**I**) Irregular linear enhancement is visible on gadolinium-enhanced T1-weighted MRI. (**J**–**L**) Postoperative MRI confirms complete resection of the lesion. (**M**–**O**) Three months of postoperative follow-up; MRI shows no recurrence.

**Figure 2 brainsci-13-00786-f002:**
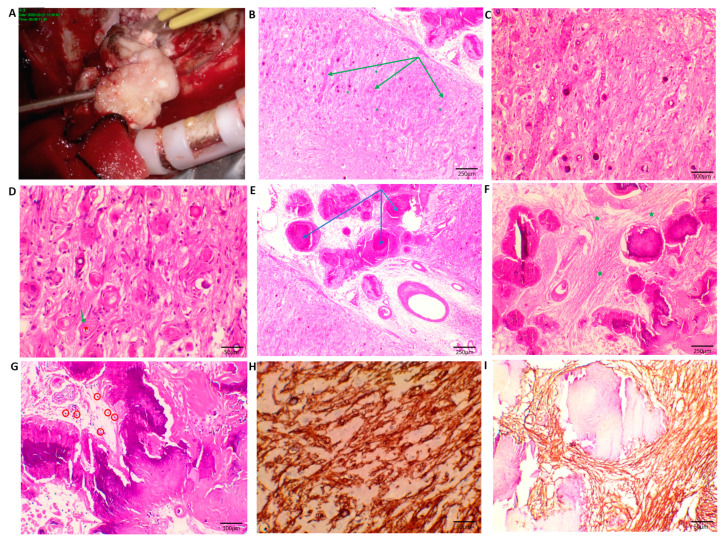
(**A**) Intraoperative microscopic photograph. (**B**) MA area within the superficial cortex of the brain parenchyma with proliferating small blood vessels (green arrows) and perivascular spindle cells (green asterisks) (H&E stain, original magnification ×40, scale bar: 1 cm = 250 μm). (**C**) Vascular proliferation, spindle cells, and psammoma bodies are visible (H&E stain, original magnification ×100, scale bar: 1 cm = 100 μm). (**D**) Proliferating blood vessels (green arrows) and psammoma bodies encased by blood vessels (red asterisks), possibly stimulated to form due to vascular proliferation (H&E stain, original magnification ×200 scale bar: 1 cm = 50 μm). (**E**) Large calcifications in the pia mater (blue arrows) (H&E stain, original magnification ×40, scale bar: 1 cm = 250 μm). (**F**) Large irregular calcifications and surrounding proliferating spindle cells (green asterisks) can be seen (H&E stain, original magnification ×40, scale bar: 1 cm = 250 μm). (**G**) Obvious calcification and some lymphocytes are visible (in red circle) (H&E stain, original magnification ×100, scale bar: 1 cm = 100 μm). (**H**) Meningioangiomatosis area: Spindle cells from perivascular hyperplasia were positive for CD34 (immunohistochemical stain, ×200, scale bar: 1 cm = 50 μm). (**I**) Calcifying pseudoneoplasms of the neuraxis area: Spindle cells around calcification are vimentin-positive (immunohistochemical stain, ×200, scale bar: 1 cm = 50 μm).

**Table 1 brainsci-13-00786-t001:** Summary of clinical features and surgical outcomes of 62 MA cases in the literature of PubMed in the last ten years.

Authors	Case	Age (yr)/Sex	Site	Clinical Presentation	Combine with	Treatment	Outcome
Austin Wheeler	1	2/male	Multiple intracranial lesions; light posterior fossa; bilateral basal ganglia; left temporal lobe; right frontal lobe, etc.	Unsteady gait	Cerebellar ependymoma	Surgery	No recurrence/alive
Omron Hassan	1	11/male	Left temporal lobe	Diplopia; headache	Meningioma(WHO II)	Surgery	No recurrence/alive
Mina S. Makary	1	17/female	Left temporal lobe	seizures; headache; dizziness	-	Surgery	No recurrence/alive; seizure-free
Brian Y.L. Chan	1	2/male	Right temporal lobe; right frontal lobe	Squint; ptosis	Arachnoid cyst	Surgery	No recurrence/alive
Kunle Oyedokun	1	3/male	Left parietal region	Seizures	-	Surgery	No recurrence/alive; seizure-free
Luke Galloway	1	18-month old/male	Right temporal lobe	Seizures	Meningioma (WHO II)	Surgery	No recurrence/alive; seizure-free
Salvatore Stilo	1	55/male	Left parietal region; head of the caudate nucleus, putamen, and thalamus	Seizures; memory and verbal impairment	B-cell central nervous system lymphoma	Surgery	NA
Alexandre Roux	1	11/male	Left frontal lobe	Seizures	-	Surgery	No recurrence/alive; seizure-free
Laura Lavalle	1	13/male	Right frontal	Stumble upon	-	Surgery	No recurrence/alive
Sara Free	2	45/male	Left occipital region	Visual disturbances; Headache	-	Topiramate	Improvement/alive
		19/male	Left occipital lobe	Epilepsy history; headache; blurring of vision	-	Observation	Improvement/alive
J. Bryan Iorgulescu	3	31/male	Frontal lobe	headache; nausea; vomiting; personality changes	Solitary fibrous tumour/hemangiopericytoma	Surgery	NA
		4/female	Parietal lobe	Seizures	Atypical teratoid/rhabdoid tumour	Surgery	No recurrence/alive; seizure-free
		2/male	Parietal lobe	Seizures	Rhabdomyosarcoma	Surgery and chemoradiotherapy	No recurrence/alive; seizure-free
Sabrina Rossi	1	6/female	Right occipital lobe; right temporo-parietal region	Seizures; intellectual disability; dysfunctional behavior	Atypical teratoid/rhabdoid tumor	Surgery	Recurrence 2 months after operation; died 14 months from first diagnosis
Raja Anand	3	6/female	Left frontal lobe	Blank stares; whole-body stiffening and rolling of the eyes	-	Surgery	One revision seven months after initial resection
		16/female	Right frontal lobe	Seizures	-	Surgery	No recurrence/alive; seizure-free
		2/male	Right parietal lobe	Seizures	-	Surgery	No recurrence/alive; seizure-free
Fábio A Nascimento	1	25/male	Left parietal lobe	Seizures	-	Surgery	NA
Shlomit Yust-Katz	1	67/male	Bilateral occipital lobes; right temporal lobe	Visual impairment	-	bevacizumab	Blind and clinically stable
Dorna Motevalli	1	13/male	Frontal lobe	Seizures	-	Surgery	No recurrence/alive; seizure-free
Daniel Joseph Donovan	1	16/female	Right temporal lobe	Seizures	-	Limited resection	No further growth/alive; seizure-free
Elif Bulut	1	55/female	Left cerebellum	Vertigo	-	Surgery	No recurrence/alive
Zhihua Sun	3	73/female	Left temporal lobe	Binocular diplopia; limited abduction of the left eye	-	Surgery	NA
		23/male	Light temporal lobe	Left hemianesthesia	-	Surgery	NA
		9/female	Left parietal lobe	Seizures	-	Surgery	NA
Y. Fu	1	23/male	Right insular lobe	Seizures	-	Surgery	No recurrence/alive
Chao Zhang	14	30/male	Right frontal lobe	Headache	Meningioma	Surgery	No recurrence/alive; seizure-free
		32/male	Right temporal lobe	Seizures	Meningioma	Surgery	No recurrence/alive; seizure-free
		3/male	Corpus callosum	Seizures	Meningioma	Surgery	No recurrence/alive; seizure-free
		12/male	Left parietal lobe	Seizures	Meningioma	Surgery	No recurrence/alive; seizure-free
		23/male	Right parietal lobe	Seizures	Meningioma	Surgery	No recurrence/alive; seizure-free
		13/male	Third ventricle	Diabetes insipidus	Meningioma	Surgery	Dead
		23/male	Right temporal lobe	Seizures	Meningioma	Surgery	No recurrence/alive; seizure-free
		10/female	Right frontal lobe	Seizures	-	Surgery	No recurrence/alive; seizure-free
		25/female	Right temporal lobe	Seizures	-	Surgery	No recurrence/alive; seizure recurrence
		26/female	Right parietal lobe	Seizures	-	Surgery	No recurrence/alive; seizure improved
		5/female	Right temporal lobe	Seizures	-	Surgery	No recurrence/alive; seizure-free
		10/male	Right parietal lobe	Seizures	-	Surgery	No recurrence/alive; seizure-free
		3.5/male	Anterior cranial fossa	Seizures	-	Surgery	No recurrence/alive; seizure-free
		27/male	Right temporal lobe	Seizures	-	Surgery	No recurrence/alive; seizure improved
Peifeng Li	1	21/female	Right temporal lobe	Seizures	-	Surgery	No recurrence/alive; seizure-free
Nobutaka Mukae	2	17/male	Left frontal lobe	Seizures	-	Surgery	No recurrence/alive; seizure-free
		16/male	Right frontal lobe	Seizures	-	Surgery	No recurrence/alive; seizure-free
A. Abdulazim	1	41/male	Right frontoparietal lobe	Monoparesis of the left leg	-	Surgery	No recurrence/alive
Ayush Batra	1	23/male	Right frontal lobe	Seizures; migraine headaches	-	Surgery	No recurrence/alive; seizure-free
Rui Feng	10	18/male	Right frontal lobe	Seizures	-	Surgery	No recurrence/alive; Engel I seizure-free
		18/male	Right frontal lobe	Seizures	-	Surgery	No recurrence/alive; Engel I seizure-free
		13/female	Left parietal lobe	Seizures	-	Surgery	No recurrence/alive; Engel I seizure-free
		39/female	Right temporal lobe	Seizures	-	Surgery	No recurrence/alive; Engel II seizure improved
		8/male	Right frontal lobe	Seizures	-	Surgery	No recurrence/alive; Engel I seizure-free
		21/male	Left parietal lobe	Seizures	-	Surgery	No recurrence/alive; Engel I seizure-free
		14/male	Left frontal lobe	Seizures	-	Surgery	No recurrence/alive; Engel III seizure improved
		17/female	Right temporal lobe	Seizures	-	Surgery	No recurrence/alive; Engel II seizure improved
		34/female	Left occipital lobe	Seizures	-	Surgery	No recurrence/alive; Engel I seizure-free
		13/male	Right parietal lobe	Seizures	-	Surgery	No recurrence/alive; Engel I seizure-free
Sara Marzi	1	37/male	Right frontal lobe	Headache	-	Surgery	No recurrence/alive
Osama Jamil	1	3/female	Left frontotemporal; left gyrus rectus	Seizures	Meningioma	Surgery	No recurrence/alive; seizure-free
Everton Barbosa-Silva	1	32/male	Right frontal lobe; Right parietal lobe; Right occipital lobe	Seizures	-	Sedation; anti-epileptic drugs	Recurrent seizures/dead
T. C. Yasha	1	19/male	Left temporal lobe	Headache; Seizures	-	Surgery	NA
Huajuan Cui	1	33/male	Left temporal lobe	Seizures	Meningioma	Surgery	No recurrence/alive; seizure-free
Katrien Jansen	1	8-month-old/male	Right temporal lobe	Seizures	-	Surgery	No recurrence/alive; seizure-free

**Table 2 brainsci-13-00786-t002:** Summary of clinical features and surgical outcomes of 49 CAPNON cases in the literature of PubMed in the last ten years.

Authors	Case	Age/Sex	Site	Presentation	Treatment	Outcome	EMA	Vimentin	S-100	GFAP
Jiri Soukup	5	38/F	Intracranial; supratentorial; central sulcus	Seizures	Surgery	No recurrence/alive	+	+	-	-
		72/F	Intracranial; supratentorial; falx cerebri	Right-sided hemiparesis; Organic psychosyndrome	Surgery	Died	+	+	-	-
		68/F	Intracranial; supratentorial; right lateral ventricle	Headaches; fainting; hydrocephalus with organic psychosyndrom	Surgery	No recurrence/alive	+	+	-	-
		50/F	Intracranial; subtentorial; right cerebellar hemisphere; and vermis	-	Surgery	No recurrence/alive	+	+	-	-
		53/F	Intracranial; subtentorial; partially intraaxial; pons and pontocerebellar angle	Headache; facial nerve palsy; tinnitus; fainting	Surgery	No recurrence/alive	+	+	-	-
Colin A. Dallimore	1	53/F	Intracranial; subtentorial; posterior fossa	Headache	Surgery	No recurrence/alive	-	NA	-	-
Wei-Qing Li	1	56/F	Intracranial; supratentorial; right frontal lobe	Headache	Surgery	No recurrence/alive	+	+	-	-
Jian-Qiang Lu	1	51/F	Spinal; paravertebral fascia in the midline at the levels of L3–4 vertebral bodies	Lower back pain; lower back mass	Surgery	NA	NA	NA	NA	NA
Lei Yan	1	44/F	Intracranial; subtentorial; skull base	Headache	Surgery	NA	-	NA	-	NA
John C. Benson	1	58/M	Intracranial; subtentorial; posterior fossa	Headache	Surgery	No recurrence/alive; Hydrocephalus	NA	NA	NA	NA
Yujian Li	1	19/F	Intracranial; supratentorial; right temporal	Seizures	Surgery	No recurrence/alive; Seizure-free	-	+	NA	+
Andrea Boschi	1	44/F	Spinal; right preforaminal extradural lesion	Back pain	Indomethacin	No recurrence/alive	NA	NA	NA	NA
Marian Preetham Suresh	1	63/M	Intracranial; supratentorial; posterior third ventricle	Cognitive impairment; gait disturbance	V-P shunt	No progress/alive	+	NA	NA	NA
Kaiyun Yang	2	57/M	Intracranial; subtentorial; extraaxial, right cerebellopontine angle (CPA)	Hoarseness; dysphagia; gait imbalance	Surgery	No recurrence/alive	+	NA	NA	NA
		70/M	Intracranial; supratentorial; right frontal lobe	Headache; gait difficulty, with falls; confusion and mood changes	Surgery	Headache improved	+	NA	NA	NA
Madoka Inukai	1	64/F	Intracranial; supratentorial; corpus callosum	Weakness of the left leg persisting	Surgery	No recurrence/alive; weakness improvement	NA	NA	-	-
Jiahua Huang	1	39/M	Intracranial; subtentorial; skull base	Visual disturbance; headache	Surgery	No recurrence/alive	+	NA	-	NA
Prashanth Raghu	1	NA/M	Intracranial; supratentorial; right medial temporal lobe	Seizures	Anti-epileptic drugs	EEG found normal; symptomatically better	NA	NA	NA	NA
Frederic A Vallejo	1	35/M	Intracranial; supratentorial; left-posterior temporal lobe	Seizures; headaches; vertigo	Surgery	No recurrence/alive; seizure-free	NA	+	NA	+
Pithon RFA	1	17/M	Intracranial; supratentorial; left frontal lobe	Seizures	Surgery	No recurrence/alive; seizure-free	NA	NA	NA	NA
Yuta Tanoue	1	52/M	Intracranial; supratentorial; left medial temporal lobe	Seizures	Surgery	No recurrence/alive; seizure-free	-	+	+	+
Zaman SKU	1	10/M	Intracranial; supratentorial; right thalamic	Left-sided hemiparesis with mixed movement disorder with hemiballism, choreoathetosis, and dystonia	Medication; physiotherapy	No progress/alive	NA	NA	NA	NA
A J Gauden	1	69/M	cranio-cervical junction	Neck pain	Surgery	No recurrence/alive	+	NA	NA	NA
Thakur B	1	67/F	Intracranial; subtentorial; cerebellum	Difficulty walking	Surgery	No recurrence/alive	-	NA	NA	+
Eric S Nussbaum	1	39/F	Intracranial–extradural; subtentorial	Right-sided deafness and tinnitus	Surgery	No recurrence/alive	NA	NA	NA	NA
Akira Watanabe	1	40/F	Intracranial; supratentorial; right frontal lobe	Somnolence	Surgery	Recurrence (after 14 months)	NA	NA	NA	NA
Atin Saha	1	67/M	Spinal; vertebral canal	Left lower extremity pain; weakness and gait instability	Surgery	No recurrence/alive; symptom improvement	NA	NA	NA	NA
Zerehpoosh FB	1	25/M	Intracranial; supratentorial; left temporal lobe	Incidental finding	Surgery	No recurrence/alive	NA	NA	NA	NA
Sean M Barber	1	31/F	Intracranial; supratentorial; right temporal lobe	Seizures	Surgery	No recurrence/alive; symptom improvement	+	+	-	-
Michael M Safaee	1	8/M	Intracranial; supratentorial; right frontal lobe	Seizures	Surgery	No recurrence/alive; seizure-free	NA	NA	NA	NA
Timothy C Blood	1	65/F	Intracranial; supratentorial; anterior cranial fossa	Deafness	Surgery	No recurrence/alive	+	NA	NA	NA
Michael A Paolini	1	17/M	Intracranial; supratentorial; left occipitoparietal lobe	Seizures	Surgery	No recurrence/alive	+	NA	NA	NA
Brasiliense LB	1	67/F	Intracranial; subtentorial; ventral midbrain and supratentorial; left frontal lobe	Seizures	Surgery	No recurrence/alive; seizure-free	NA	NA	-	-
Abdaljaleel M	1	62/F	Intracranial; supratentorial; temporal, parietal, and occipital lobes	Seizures; headaches	Surgery	NA	NA	NA	-	NA
Nayuta Higa	1	62/M	Intracranial; supratentorial; left cingulate gyrus	Headache	Surgery	Recurrence	NA	NA	+	+
Hong Gang Wu	1	39/F	Spine; sacral canal	Sacrococcygeal pain	Surgery	No recurrence/alive	NA	NA	NA	NA
Sara García Duque	4	48/F	Intracranial; supratentorial; left occipital lobe	Headache	Surgery	No recurrence/alive	NA	NA	NA	NA
		51/F	Spinal cord; L2	Lower back pain	Surgery	No recurrence/alive	NA	NA	NA	NA
		46/F	Spinal cord; C3	Posterior neck pain	Surgery	No recurrence/alive	NA	NA	NA	NA
		73/M	Spinal cord; T2	Progressive paraparesis	Surgery	No recurrence/alive	NA	NA	NA	NA
Joseph Ghaemi	1	18/M	Intracranial; subtentorial; skull base	Headache; diplopia	Surgery	NA	+	NA	NA	NA
Arthur J M Lopes	1	72/F	spinal cord; L2	Low back pain	Surgery	No recurrence/alive	+	NA	+	+
Mohammed Alshareef	1	59/F	cranio-cervical junction	Gait instability; balance difficulty	Surgery	No recurrence/alive	NA	NA	NA	NA
Molly Hubbard	1	38/F	Intracranial; supratentorial; bilateral frontal lobe	Headache; left facial numbness	Surgery	No recurrence/alive; symptom improvement	NA	NA	NA	NA
Karol Wiśniewski	1	29/M	Intracranial; subtentorial; foramen magnum	Headache	Surgery	No recurrence/alive	+	NA	-	NA
Kirill Lyapichev	1	24/M	Intracranial; supratentorial; right temporo-occipital lobe	Headache; seizures; loss of vision	Surgery	No recurrence/alive	-	NA	NA	+
M N Stienen	2	46/M	Intracranial; supratentorial; right parietal lobe	Seizures	Surgery	No recurrence/alive	NA	NA	NA	NA
		55/F	Intracranial; supratentorial; left frontoparietal lobe	Progressive hallucinosis; behavioral disorders	Surgery	No recurrence/alive; symptom improvement	NA	NA	NA	NA
Edward E Kerr	1	56/M	Intracranial; subtentorial; posterior fossa	Headache	Surgery	No recurrence/alive; symptom improvement	+	NA	NA	-
Mun Keong Kwan	1	48/M	Spinal, extradural mass located dorsal to the T9–T10 disc	Radicular pain	Indomethacin	No recurrence/alive	NA	NA	NA	NA

## Data Availability

Data are available upon request.

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
