# Peer review of "Meningioangiomatosis Combined with Calcifying Pseudoneoplasms of Neuraxis"

_brainsci, 2023, doi:10.3390/brainsci13050786_

Round 1

Reviewer 1 Report

The manuscript can be accepted after the authors correct the following comments:

1.     Is this article a case report or a literature, because the authors are mentioned in the title both of them. Thus, kindly, I think in my opinion remove the literature because this is a case report.

2.     The abstract is consistent and written in a good way. However, it needs a native English speaker.

  1. Please note that the introduction part needs further paragraphs. Kindly, add a second short paragraph at the introduction section with modern references.

4.     Please make double check about the academic writing (needs native speakers in English) for whole manuscript.

5.     The method, result, and discussion sections are written and explained well.

  1. The manuscript is lacked the conclusion section. Please write it to reflect the results in a good way.

  1. Make sure that all sentences are linked together.

Dear Editor in chief,

Greeting,

 Please ensure that the entire manuscript is edited by the authors and someone with proficiency in native English.

Thank you

Author Response

Dear reviewer:

Thank you for your decision and constructive comments on my manuscript. We have carefully considered the suggestion of Reviewer and make some changes. We have tried our best to improve and made some changes in the manuscript. Please see the attachment.

Reviewer 2 Report

Meningiomatosis combined with calcifying pseudoneoplasms of neuraxis: A Case Report and Literature Review the case report is well written and really interesting. 

The major flaws are the organization of the sections and the introduction and method that must be developped. 

Introduction: Please develop the introduction by giving more background and the aims and hypothesis for this paper. 

Method: the method should be in the results section. The methods section should describe the population, the materials and methods used for the study. Did you get an informed consent or ethic committee approval?

Results Figure 1: In the text please put the letter of the image when citing in capital because at every other places they are in capital form .

Figure 2: On the images there is a scale bar, please add in the legend what is the corresponding length of the scale bar.

Discussion: Please develop the perspectives, what does it bring to the community? It would be interesting to discuss how to improve the diagnosis 

Author Response

(The authors gave the same response as above.)

Reviewer 3 Report

Authors present a first case report on a patient who presented with combined meningioangiomatosis and calcyfiyng pseudoneoplasm of the neuraxis (CAPNON) in a 31 years old female patient with left frontal calcified lesion with OCD who underwent surgery. Manuscript is fairly written and case presentation provides enough information on the case. Since these are very rare lesions, there is no reason not to expand the literature review with cases which are older than 10 years; I suggest to include reviews and comment:

Stienen MN, Abdulazim A, Gautschi OP, Schneiderhan TM, Hildebrandt G, Lücke S. Calcifying pseudoneoplasms of the neuraxis (CAPNON): clinical features and therapeutic options. Acta Neurochir (Wien). 2013 Jan;155(1):9-17. doi: 10.1007/s00701-012-1502-2. Epub 2012 Oct 3. PMID: 23053277.

Furthermore, literature review is not complete, some new series are missing:

Domecq Laplace L, Ruella M, Caffaratti G, Villamil F, Monsalve M, Alcorta SC, Cervio A. Posterior Fossa Calcifying Pseudoneoplasm of the Neuraxis (CAPNON): Presentation of Three Surgical Cases. World Neurosurg. 2022 Nov;167:e423-e431. doi: 10.1016/j.wneu.2022.08.022. Epub 2022 Aug 11. PMID: 35964906.

Consider including an operative video. 

Acceptable. 

Author Response

(The authors gave the same response as above.)

Round 2

Reviewer 2 Report

thanks for the changes